# Oxygen Vacancies-Rich S-Cheme BiOBr/CdS Heterojunction with Synergetic Effect for Highly Efficient Light Emitting Diode-Driven Pollutants Degradation

**DOI:** 10.3390/nano13050830

**Published:** 2023-02-23

**Authors:** Yang Yu, Fengjuan Chen, Xuekun Jin, Junyong Min, Haiming Duan, Jin Li, Zhaofeng Wu, Biaobing Cao

**Affiliations:** 1Key Laboratory of Solid State Physics and Devices Autonomous Region, School of Physics Science and Technology, Xinjiang University, Urumqi 830046, China; 2Key Laboratory of Energy Materials Chemistry, Ministry of Education, Institute of Applied Chemistry, Xinjiang University, Urumqi 830046, China; 3School of Environment, Tsinghua University, Beijing 100084, China

**Keywords:** BiOBr, CdS, oxygen vacancy, S-scheme, photocatalysis

## Abstract

Recently, the use of semiconductor-based photocatalytic technology as an effective way to mitigate the environmental crisis attracted considerable interest. Here, the S-scheme BiOBr/CdS heterojunction with abundant oxygen vacancies (Vo-BiOBr/CdS) was prepared by the solvothermal method using ethylene glycol as a solvent. The photocatalytic activity of the heterojunction was investigated by degrading rhodamine B (RhB) and methylene blue (MB) under 5 W light-emitting diode (LED) light. Notably, the degradation rate of RhB and MB reached 97% and 93% in 60 min, respectively, which were better than that of BiOBr, CdS, and BiOBr/CdS. It was due to the construction of the heterojunction and the introduction of Vo, which facilitated the spatial separation of carriers and enhanced the visible-light harvest. The radical trapping experiment suggested that superoxide radicals (·O_2_^−^) acted as the main active species. Based on valence balance spectra, Mott-Schottky(M-S) spectra, and DFT theoretical calculations, the photocatalytic mechanism of the S-scheme heterojunction was proposed. This research provides a novel strategy for designing efficient photocatalysts by constructing S-scheme heterojunctions and introducing oxygen vacancies for solving environmental pollution.

## 1. Introduction

Nowadays, the environmental deterioration caused by rapid socioeconomic development is becoming increasingly serious [1,2,3]. Photocatalysis as a green technology can effectively solve these problems [4,5,6]. Semiconductor-based photocatalysts can degrade pollutants and transform CO_2_, and N_2_ into high-value products through sunlight [7,8,9]. So far, many materials have attracted focus due to their considerable photocatalytic properties, such as TiO_2_, CdS, ZnO, SnO_2_, WO_3_, etc [10,11,12,13,14]. As a promising layered photocatalyst, BiOBr has been extensively investigated in the photocatalytic area because of its good stability. However, the charge carriers in BiOBr are easy to recombine and the light utilization of BiOBr is low. Therefore, constructing a high-efficiency BiOBr-based photocatalyst is still a challenge.

Oxygen vacancies (Vo) in metal oxide semiconductors are significant for improving photocatalytic performance [15,16,17] because introducing Vo can narrow the band gap of photocatalysts [18]. Hence, introducing oxygen vacancies (Vo) can expand the visible light absorption range to enhance the photocatalytic performance [19,20,21]. For example, Chang et al. successfully synthesized Bi_2_O_2_CO_3_ with oxygen vacancies (Vo-BOC) by a precipitation method. The bandgap energy of BOC was 2.65 eV, and that of Vo-BOC was 2.22 eV [22]. In addition, oxygen vacancies gave rise to active sites and increased the density of local electrons to promote surface reactions. For instance, Zhao et al. prepared Vo-BiOBr via the solvothermal method. The oxygen vacancy-rich BiOBr showed higher photocatalytic activity than pristine BiOBr. The ability of O to trap electrons from Vo led to an increase in electron densities of O in Vo-BiOBr compared with BiOBr [23]. However, we hope to further improve the photocatalytic performance of Vo-BiOBr.

Research showed that compounding narrow bandgap semiconductors can effectively improve photocatalytic performance. As an important metal sulfide, CdS with a suitable bandgap of 2.4 eV is broadly applied for degrading dyes and antibiotics [24,25]. However, charge carriers are prone to recombination, which restricts their practical application [26]. Constructing heterojunctions with other semiconductors is an effective method to overcome the disadvantages of CdS. Wang et al. synthesized the CdS/WS_2_ heterojunction by a two-step hydrothermal method, and 88% of RhB was degraded in 40 min [27]. The bandgap structure of Vo-BiOBr matches well with that of CdS. Hence, CdS is suitable for the construction of a heterojunction with BiOBr to facilitate the separation of charge carriers. Several studies on BiOBr/CdS heterojunctions were reported in recent years [28,29,30]. However, little was reported in the studies on the synergistic effect of Vo and S-scheme heterojunctions to enhance photocatalytic activity.

Inspired by the studies mentioned above, we have synthesized BiOBr/CdS photocatalysts with rich oxygen vacancies using a solvothermal approach. The photocatalytic properties of Vo-BiOBr/CdS composites were measured by degrading RhB and MB under an LED light. Photocurrent and electrochemical impedance (EIS) tests were carried out to measure the separation ability of charge carriers. Moreover, the S-scheme mechanism was proposed through a trapping experiment, band structure, and work function. This study could offer a new strategy to prepare efficient photocatalysts.

## 2. Materials and Methods

### 2.1. Materials

Ethylene glycol (EG), cadmium nitrate tetrahydrate (Cd(NO_3_)_2_·4H_2_O), L-Cysteine, ethylenediamine, tetracycline (TC), and potassium dichromate(K_2_Cr_2_O_7_) were obtained from Shanghai Aladdin Biochemical Technology Co., Ltd. (analytically pure, 99%, Shanghai, China). Polyvinyl pyrrolidone (PVP), isopropanol, potassium iodide, and p-benzoquinone were obtained from the Macklin company (analytically pure, 99%, Shanghai, China). Bismuth nitrate pentahydrate (Bi(NO_3_)_3_·5H_2_O), potassium bromide (KBr), MB, and RhB was purchased from Tianjin Kwangfu Fine Chemical Research Institute (analytically pure, 99%, Tianjin, China).

### 2.2. Synthesis

Vo-BiOBr/CdS heterojunction was constructed by a two-step solvothermal method. Firstly, 2.01 g of Cd(NO_3_)_2_·4H_2_O and 2.04 g of L-Cysteine were put into a beaker containing 30 mL ethylenediamine and stirred for 1 h. Then the above dispersion was transferred into a 100 mL of Teflon-lined stainless-steel autoclave and heated at 180 °C for 5 h. The resulting yellow product was washed with water and ethanol three times and dried at 60 °C to obtain CdS.

Secondly, 0.485 g of Bi(NO_3_)_3_·5H_2_O was dispersed in a 30 mL solution containing 5 mL EG and 25 mL H_2_O. Then, 0.4 g of PVP was added to the above solution. Next, CdS nanorods (0.03 g) and 0.12 g of KBr were added to the above-mentioned solution and stirred for 2 h. Afterward, the dispersion was transferred into a 100 mL Teflon-lined stainless-steel autoclave and heated at 160 °C for 12 h. The obtained sample was washed with water and ethanol three times and dried at 60 °C. Finally, Vo-BiOBr/CdS was successfully prepared (the mass ratio of Vo-BiOBr and CdS is 10:1). Similarly, other Vo-BiOBr/CdS of different mass ratios were synthesized by adjusting the mass ratio of Vo-BiOBr and CdS as 20:1, 20:3, 5:1, 4:1, 10:3, and 2:1, and labeled as Vo-BiOBr/CdS-5, Vo-BiOBr/CdS-15, Vo-BiOBr/CdS-20, Vo-BiOBr/CdS-25, Vo-BiOBr/CdS-30, Vo-BiOBr/CdS-50, respectively. For comparison, BiOBr/CdS and BiOBr were synthesized in deionized water without adding PVP and CdS, respectively.

### 2.3. Characterization

XRD pattern was obtained with a Bruker D8 Advance X-ray diffractometer using Cu-Kα radiation (λ = 1.5405 nm). The X-ray photoelectron spectroscopy (XPS, Thermo Scientific ESCALAB 250Xi) was applied to get the chemical status and valence state of samples. The transmission electron microscopy (TEM), selected area electron diffraction (SAED), and energy dispersive X-ray (EDX) mapping and spectrum were measured by an FEI Tecnai G2 F20 S-Twin with a 200 kV accelerating voltage. UV-vis diffuse reflection spectra (UV-vis DRS) were performed on a Lambda 650 UV-vis spectrophotometer at wavelengths of 200–800 nm. The photocurrent response, EIS, and M-S curves were measured on the CHI-660E electrochemical workstation.

### 2.4. Photocatalytic Test

The photocatalytic performance of the photocatalysts was assessed by the degradation of RhB (10 mg·L^−1^), and MB (20 mg·L^−1^) under visible light irradiation using a 5 W LED lamp (PerfectLight, λ > 420 nm). Typically, 20 mg of catalysts were put into 40 mL RhB (or MB) solution and stirred for 30 min in the dark. After irradiation, 4 mL of the liquid was sampled and centrifuged. The concentration of the solution was characterized using a UV-vis spectrometer.

To study the ability of Vo-BiOBr/CdS to degrade other pollutants, the degradation of tetracycline (TC, 20 mg·L^−1^) and reduction of potassium dichromate (Cr^6+^, 20 mg·L^−1^) were carried out. To study the ability of Vo-BiOBr/CdS to degrade multiple pollutants simultaneously, the degradation and reduction of mixed solution with a volume of 1:1:1 containing RhB (10 mg·L^−1^), MB (10 mg·L^−1^), and Cr^6+^ (20 mg·L^−1^) were conducted.

The photocatalytic stability of the samples can be tested by cycling experiments of photocatalytic degradation of RhB. After one degradation, the samples were filtered, washed, and dried before repeating the degradation experiment.

To investigate the photocatalytic species, 5 mL 0.1 mmol/L radical scavengers (isopropanol, 1,4-benzoquinone, and KI) were added into the RhB solution in the above photocatalytic process.

### 2.5. Theoretical Calculations

First-principle calculations based on Density functional theory (DFT) were performed using the VASP software Package with plane wave techniques [31]. Perdew–Burke–Ernzerhof (PBE) [32] functional framework was used to describe the exchange-correlation functional interaction. The truncation energy of the Plane Wave Truncation Base Group was 500 eV. The energy band structures and density of states (DOS) of the 2 × 2 × 1 BiOBr and Vo-BiOBr supercell model and the hexagonal CdS cell model were calculated respectively (Appendix A).

## 3. Results

### 3.1. Structure and Morphology

Firstly, the CdS nanorods were synthesized via a solvothermal approach (Figure 1). Secondly, the Vo-BiOBr/CdS was prepared by dispersing the CdS in EG-H_2_O solution containing PVP, KBr, and Bi(NO_3_)_3_. The oxygen vacancies were introduced by adding ethylene glycol (EG) to the solution [33]. The addition of PVP facilitated the creation of Vo because it tends to bind with the unsaturated Bi atom on the BiOBr surface to decrease the surface energy, which leads to the creation of more Vo [34]. In addition, PVP as a common capping agent can also affect the morphology of the catalyst [35].

The crystal structures and phases of BiOBr, Vo-BiOBr, CdS, and Vo-BiOBr/CdS were identified by XRD measurement (Figure 2a) [36]. It could be observed that the main diffraction peaks at 10.90°, 25.15°, 31.69°, 32.22°, 46.20°, and 57.11° corresponded to the (001), (101), (102), (110), (200), and (212) planes of BiOBr and Vo-BiOBr, which can be indexed to tetragonal phase BiOBr (JCPDS No. 09-0393) [37]. Notably, diffraction peaks at 21.92° and 44.72° were not observed in BiOBr, which may be due to the introduction of oxygen defects [38]. Furthermore, the intensity of (001) and (102) planes markedly decreased, while that of (110) increased compared with BiOBr. The possible reason was that when the relative intensity of (110)/(001) planes increased, the structure of Vo-BiOBr tended to form a hierarchical structure [39], which can be verified by BiOBr nanoplate with an exposed (001) facet and Vo-BiOBr nanosheets with a (110) facet. The (002), (100), (101), (110), and (112) planes of CdS and the (001), (200), and (211) planes of Vo-BiOBr were found in the Vo-BiOBr/CdS (Figure 2b,c), which proved successful compound of Vo-BiOBr and CdS [30].

The morphology of the photocatalysts was studied by TEM technology. Figure 3a,b showed Vo-BiOBr as nanosheets and CdS as nanorods with a diameter of tens of nanometers. In Figure 3c,d, CdS nanorods were distributed on the surface of Vo-BiOBr nanosheets. According to the HRTEM image of Vo-BiOBr/CdS (Figure 3e), the lattice spacing of 0.281 nm assigned to the (110) plane of Vo-BiOBr and 0.332 nm assigned to the (002) plane of CdS was observed [40,41]. Furthermore, a distinct interface could be found, demonstrating the construction of a heterojunction between Vo-BiOBr and CdS [42]. The obscured, circled lattice fringes were observed (Figure 3e), meaning that defects existed in Vo-BiOBr [43]. In addition, the SAED pattern of Vo-BiOBr/CdS (Figure 3f) showed clear diffraction rings, indicating that Vo-BiOBr/CdS was polycrystalline. The diffraction rings can be ascribed to the (110), (101) planes of Vo-BiOBr and the (110), (112) planes of CdS. The elemental mapping images (Figure 3g) revealed the presence of Cd, S, Bi, O, and Br elements, which were evenly distributed. The EDX spectrum (Figure 3h) displayed the characteristic peaks of Bi, O, Br, S, and Cd elements, indicating that Vo-BiOBr/CdS was synthesized.

The chemical composition of as-synthesized samples was evaluated by XPS measurement. Survey spectra from Figure 4a of Vo-BiOBr, CdS, and Vo-BiOBr/CdS demonstrate the presence of Bi, O, Br, S, and Cd elements in Vo-BiOBr/CdS. The high-resolution XPS spectra of Bi 4f, Br 3d, O 1s, Cd 3d, and S 2p for the samples are presented in Figure 4b–f. The spectra of Bi 5f exhibited two peaks at 158.78 eV and 164.08 eV (Figure 4b), which can be ascribed to Bi 4f_7/2_ and Bi 4f_5/2_, respectively, demonstrating the presence of Bi^3+^. As for Br 3d (Figure 4c), two peaks located at 67.88 eV and 68.93 eV, can be ascribed to Br 3d_5/2_ and Br 3d_3/2_, respectively, indicating the chemical states of Br^−^. In Figure 4d, the spectra of O 1s can be fitted with three peaks at 529.38 eV, 530.83 eV, and 532.33 eV inferring lattice oxygen (Bi-O), oxygen vacancies, and absorbed oxygen, respectively [44]. It is worth noting that Vo-BiOBr/CdS had a larger area for the characteristic peak representing the oxygen vacancy than BiOBr (Appendix A), suggesting that the reduction of ethylene glycol is vital to forming Vo. More significantly, the peaks of Bi 4f, Br 3d, and O 1s of Vo-BiOBr moved toward lower binding energy compared with BiOBr (Appendix A), which proved that the charge density increased because of the local electrons of oxygen vacancies. In Figure 4e, the peaks at 404.68 eV and 411.43 eV matched Cd 3d_5/2_ and Cd 3d_3/2_, respectively; the other peaks originated from satellite peaks of Cd^2+^. The peaks at 161.08 eV and 162.48 eV were ascribed to S 2p_3/2_ and S 2p_1/2_ (Figure 4f), respectively [45]. Notably, the Bi 4f, Br 3d, Cd 3d, S 2p, and O 1s peaks of Vo-BiOBr/CdS shifted to higher binding energy compared with Vo-BiOBr or CdS, which may be due to the changing of the surface-electron density, suggesting the formation of a heterojunction in Vo-BiOBr/CdS. Consequently, when CdS coupled with BiOBr, the transfer channel of carriers would be constructed, causing a charge redistribution between Vo-BiOBr and CdS. This facilitated the charge transfer and enhanced the photocatalytic activity.

### 3.2. Optical Property

The optical performance of as-synthesized samples was examined by UV-vis DRS (Figure 5a). Vo-BiOBr exhibited a stronger absorption of ultraviolet and visible light than that of BiOBr, and the increased absorption intensity in 550–700 nm was evident. This was attributed to the introduction of Vo, which enhanced the absorption intensity of visible light [46]. When coupling with CdS, Vo-BiOBr/CdS showed an enhanced absorption intensity and an occurrence of a red shift compared with Vo-BiOBr. It indicated that oxygen vacancies and the heterojunction greatly enhanced the visible light absorption ability.

The bandgap energies (Eg) of samples could be identified by the following formula (αhυ)*n*= A(hυ − E_g_) (Here, *n* = 2 for direct semiconductors and *n* =1/2 for indirect semiconductors) [47]. As for Vo-BiOBr and CdS, the *n* values were 1/2 and 2, respectively [48]. Hence, the bandgap energies of BiOBr, Vo-BiOBr, and CdS were 2.78, 2.14, and 2.40 eV, respectively (Figure 5b). Notably, the E_g_ of Vo-BiOBr was smaller than that of BiOBr, suggesting that oxygen vacancy could narrow the band gap energy [49], which is helpful for the utilization of visible light.

### 3.3. Enhanced Photocatalytic Performance

The photocatalytic activity of Vo-BiOBr/CdS was evaluated by the degradation of dyes under LED light irradiation. The photocatalytic process started when the dye molecules were adsorbed onto the photocatalyst surface in the dark to achieve equilibrium. In Figure 6a, the concentration of RhB had no significant change in the absence of photocatalysts, indicating high stability. The CdS and pristine BiOBr without oxygen vacancies exhibited low photodegradation efficiency: 42% and 79% RhB were decomposed after 60 min, respectively. When the BiOBr was coupled to CdS nanorods, the composites (BiOBr/CdS) exhibited slightly enhanced photodegradation efficiency, owing to the construction of heterojunction, which promoted the fast separation of photogenerated carriers [50]. Compared with BiOBr, CdS, and BiOBr/CdS, the Vo-BiOBr/CdS heterojunction displayed the highest photocatalytic performance when the mass ratio of CdS to Vo-BiOBr was 0.1 (Appendix A). Nearly 97% of RhB was degraded when irradiated for 60 min under LED light, suggesting that the synergistic effect of Vo and the heterojunction played a crucial part in promoting photocatalytic activity. Under similar conditions, the Vo-BiOBr/CdS displayed a superior photocatalytic performance to that of the reported (Appendix A).

To briefly represent the photocatalytic degradation rate toward RhB of samples, the first-order model (ln (C_o_/C) = kt) was adopted [51], and the fitting results were depicted in Figure 6c. The rate constant (k) of Vo-BiOBr/CdS was calculated to be 0.053 min^−1^, about 3.4, 6.1, and 2.2 times as high as those of BiOBr, CdS, and BiOBr/CdS, respectively. Similarly, Vo-BiOBr/CdS showed the highest photocatalytic properties toward the MB (Figure 6b). The degradation rate by Vo-BiOBr/CdS reached 94% after irradiation for 60 min. In Figure 6d, its corresponding k was 0.041 min^−1^, about 9.2, 9.6, and 3.9 times as high as that of BiOBr, CdS, and BiOBr/CdS, respectively. The time-dependent UV-vis spectrum of Vo-BiOBr/CdS for degrading RhB and MB under LED light was illustrated in Figure 6e,f, respectively. A gradual decrease in the characteristic peak intensity of RhB and MB could be observed as time increased, which proved that RhB and MB were degrading [46].

During the wastewater treatment process, the concentration of RhB could affect the degradation rate of photocatalysts. As depicted in Figure 7a, the impact of RhB concentrations on the degradation rate of Vo-BiOBr/CdS was also conducted. With the RhB concentration increasing, the removal rate decreased. After 60 min of irradiation, the degradation rate reached 96%, 90%, and 86% when the initial concentrations were 10 mg/L, 20 mg/L, and 30 mg/L, respectively. The impact of catalyst dosage on removal efficiency was also explored; as illustrated in Figure 7b, the degradation rate was enhanced when the dosage of Vo-BiOBr/CdS was changed from 10 mg to 20 mg. However, when it increased from 20 to 30 and 40 mg, degradation efficiencies just slightly increased. Therefore, 20 mg was the best choice considering the cost and degradation efficiency.

To investigate the degradation and reduction efficiency for pollutants other than dyes, the photodegradation of TC and photoreduction of Cr (VI) to Cr(III) were performed. The degradation and reduction rates of TC and Cr (VI) by Vo-BiOBr/CdS were 71% and 99%, respectively (Appendix A). The above results proved that the photocatalysts exhibited efficient performance for different pollutants. In addition, the actual wastewater often contained various pollutants. Hence, mixed solutions containing RhB, Cr^6+^, and MB by volume ratio 1:1:1 were employed to investigate the effect of co-existing contaminants on photodegradation and photoreduction efficiencies. As depicted in Figure 7c,d, Vo-BiOBr/CdS exhibited excellent photocatalytic activity toward the mixed solution. It is worth noting that the photodegradation rate toward RhB decreased from 97% to 82%, possibly due to competitive interactions between various pollutants [52].

The photocurrent response and EIS measurements were performed to study a better photocatalytic performance of Vo-BiOBr/CdS compared with BiOBr, CdS, and BiOBr/CdS. As illustrated in Figure 8a, the Vo-BiOBr/CdS showed higher photocurrent density compared with BiOBr/CdS, indicating that introducing Vo could facilitate the separation of charge carriers. Additionally, the photocurrent density of BiOBr/CdS was higher than that of CdS and BiOBr, demonstrating that heterostructures could also facilitate the splitting of charge carriers. Subsequently, the quantum efficiencies of photocatalysts were studied by EIS. In Figure 8b, Vo-BiOBr/CdS showed the smallest radius among the as-prepared photocatalysts in the Nyquist plots, representing smoother interface charge transfer paths and enhanced mobility [53]. Hence, the increased photocatalytic performance could attribute to the synergistic effect between the oxygen vacancies and the heterojunction, which accelerated the fast separation of charge carriers. The cycle experiments of RhB on Vo-BiOBr/CdS were performed to examine the stability of the photocatalyst (Figure 8c) and the degradation efficiency slightly decreased, indicating that Vo-BiOBr/CdS had good stability.

The primary active species in the degradation of RhB on Vo-BiOBr/CdS was discussed by radical trapping experiments. 1,4-benzoquinone (BQ), isopropanol (IPA), and potassium Iodide (KI) were used as the quenchers of superoxide radicals (·O_2_^−^), hydroxyl radicals (·OH), and holes (h^+^), respectively [54]. The dosage of each quencher was maintained equally. As depicted in Figure 8d, the removal rate for RhB significantly decreased in the presence of BQ, which indicated that ·O_2_^−^ radicals were the main active species [53]. In conclusion, the ·O_2_^−^ was the decisive active species.

### 3.4. Band Structure and Density of States by DFT Calculation

The band structure of BiOBr, Vo-BiOBr, and CdS were simulated through DFT calculation (Figure 9). The CBM and the VBM were not at the same point, suggesting that BiOBr had an indirect bandgap (Figure 9a). The CdS was a direct band gap semiconductor (Figure 9c). The band gap values of BiOBr and CdS were identified to be 2.24 eV and 1.13 eV, respectively, which was narrower than the experimental values of 2.78 eV and 2.4 eV (may be due to the shortcoming of the DFT calculation). To study the impact of oxygen vacancies, we conducted DFT theoretical calculations to obtain the calculated band structure of Vo-BiOBr (Figure 9e). It is worth noting that a new defect level was created by oxygen vacancies compared with BiOBr. The defect levels facilitate the excitation of electrons and transfer to the CB, thus reducing the band gap of BiOBr.

Moreover, we calculated the DOS of BiOBr, Vo-BiOBr, and CdS. Figure 9b shows that the CBM was mainly dominated by Bi 6p orbitals, while the VBM was occupied by O 2p and Br 4p orbitals of BiOBr. The CBM and VBM were both dominated by S 3p orbitals of CdS (Figure 9d). The DOS for Vo-BiOBr showed the presence of defect levels (Figure 9f), in agreement with the results of the band structure.

### 3.5. Photocatalytic Mechanism

As illustrated in Figure 10a,b, the Mott–Schottky (M-S) spectra of both CdS and Vo-BiOBr were positive, meaning that they are both n-type semiconductors [55]. The flat band (E_fb_) potential was measured to be −0.18 and −1.11 eV (vs NHE) respectively. The conduction band (CB) potential (E_CB_) of Vo-BiOBr and CdS were identified to be −0.18 and −1.11 eV, respectively, because the E_CB_ of the n-type semiconductor was nearly equal to E_fb_ [56]. Furthermore, the corresponding valance band potential (E_VB_) can be calculated by the formula E_VB_ = E_CB_ + E_g_ [57]. Here, E_g_ is band gap energy, identified as 2.14 and 2.4 eV for Vo-BiOBr and CdS by UV-vis DRS. Therefore, the calculated E_VB_ for Vo-BiOBr and CdS were 1.96 and 1.29 eV, respectively. The energy gap between the Fermi level (E_f_) and the VB could be identified by the valance band-XPS spectra of Vo-BiOBr and CdS (Figure 10c,d) [58], suggesting that the E_f_ of Vo-BiOBr and CdS were 0.32 and −0.16 eV, respectively.

Based on the previous analyses, the photocatalytic mechanism of Vo-BiOBr/CdS was studied. According to the results of M-S spectra, UV-vis DRS, and XPS valance band, the band structure, as illustrated in Figure 10e, shows that if the composite was a type-II heterojunction, the photoexcited electrons migrated from the CB of CdS to that of Vo-BiOBr and accumulated in the CB of Vo-BiOBr, and the photoexcited holes could migrate from the VB of Vo-BiOBr to CdS and accumulate in the VB of CdS. However, the E_CB_ of Vo-BiOBr was lower than O_2_/O_2_^−^ (−0.33 eV). As a result, the electrons in the CB of Vo-BiOBr could not reduce dissolved oxygen to ·O_2_^−^, which contradicted the hypothesis that ·O_2_^−^ was the main active species. According to the above analysis, the photocatalysts could not obey the type-II heterojunction, and the more suitable photocatalytic mechanism was the S-scheme heterojunction. Superoxide radicals had strong oxidative activity, and they could oxidize organic matter to carbon dioxide and water. The main reactions of photocatalytic degradation were as follows [59]:Vo-BiOBr/CdS + hv → Vo-BiOBr (e^−^ + h^+^)/CdS (e^−^ + h^+^)(1)
Vo-BiOBr (e^−^) + CdS (h^+^) →Vo-BiOBr/CdS (recombination)(2)
CdS (e^−^) + O_2_ →·O_2_^−^(3)
·O_2_^−^ + RhB/MB → CO_2_ + H_2_O(4)

To further identify the charge migration pathway of Vo-BiOBr/CdS heterojunction, the work functions of Vo-BiOBr and CdS were calculated. In Figure 11a,b, the electrostatic potentials of Vo-BiOBr (001) and CdS (001) planes were identified to be 3.38 eV and 3.01 eV respectively. The work function of Vo-BiOBr was larger than that of CdS, indicating that electrons could transfer from CdS to Vo-BiOBr spontaneously.

In the S-scheme heterojunction, the Vo-BiOBr presented a bigger Φ and lower E_f_ than CdS. Therefore, the S-scheme heterojunction mechanism was proposed (Figure 11c). The E_f_ of Vo-BiOBr was lower than that of CdS, so when the two semiconductors contacted each other, electrons would migrate from CdS to Vo-BiOBr until their E_f_ reached equilibrium [60]. This process resulted in the construction of an internal electric field (IEF) directing from CdS to Vo-BiOBr. Under the LED irradiation, electrons were exited from VB to CB in Vo-BiOBr and CdS, respectively. IEF promoted the recombination of photoexcited electrons in the CB of Vo-BiOBr and holes in the VB of CdS [61]. Moreover, the rest of the electrons and holes possessing strong redox potential would be retained. Therefore, the ·O_2_^−^ formation process mainly depended on electrons in the CdS on account of the S-scheme mechanism, but the ·OH could not be produced since the E_VB_ of Vo-BiOBr was higher than H_2_O/·OH (2.39 eV). In conclusion, this S-scheme heterojunction promoted charge separation and retained a strong redox capability.

## 4. Conclusions

In conclusion, the S-scheme Vo-BiOBr/CdS heterojunction was synthesized by the solvothermal approach. The obtained S-scheme Vo-BiOBr/CdS heterojunction displayed RhB and MB removal efficiencies of 97 % and 94%, respectively. Moreover, the composite showed a good degradation and reduction efficiency for RhB-Cr(VI)-MB mixed solutions. A significant enhancement of the photocatalytic activity can be ascribed to the synergistic effect of the heterojunction and Vo. The results of the radical trapping experiment, valence band-XPS, and work function proved the formation of the S-scheme heterojunction. Finally, the four cycles of RhB degradation on Vo-BiOBr/CdS proved its good stability. This study provides a novel insight into designing highly efficient photocatalysts for environmental remediation.

## Figures and Tables

**Figure 1 nanomaterials-13-00830-f001:**
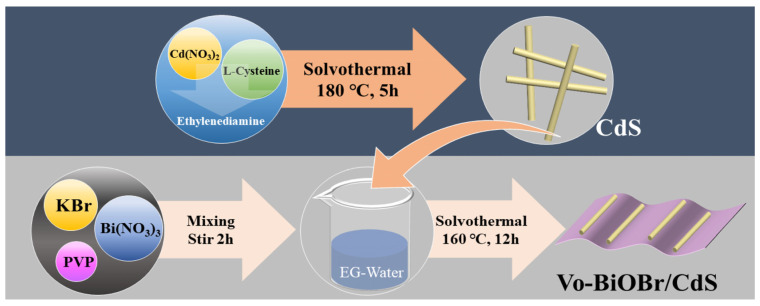
Diagram of the synthesis process of Vo-BiOBr/CdS photocatalysts.

**Figure 2 nanomaterials-13-00830-f002:**
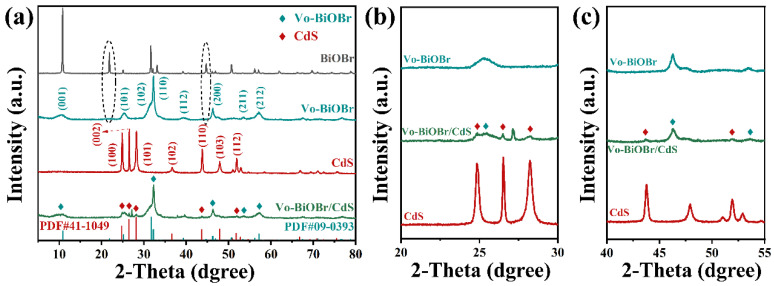
XRD patterns of Vo-BiOBr/CdS in different ranges of diffraction angle: (**a**) 5–80°, (**b**) 20–30°, (**c**) 40–55°.

**Figure 3 nanomaterials-13-00830-f003:**
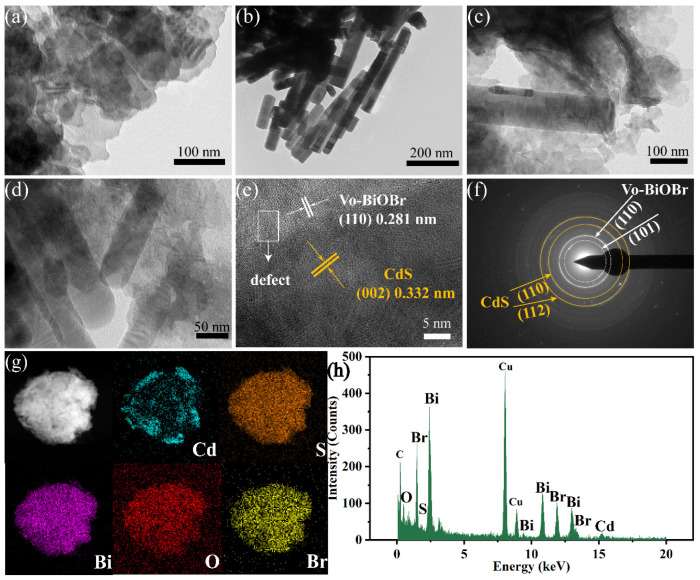
TEM images of Vo-BiOBr (**a**), CdS (**b**), and Vo-BiOBr/CdS (**c**,**d**); HRTEM image (**e**); SAED pattern (**f**); The elemental mapping (**g**), and EDX spectrum (**h**) of the Vo-BiOBr/CdS.

**Figure 4 nanomaterials-13-00830-f004:**
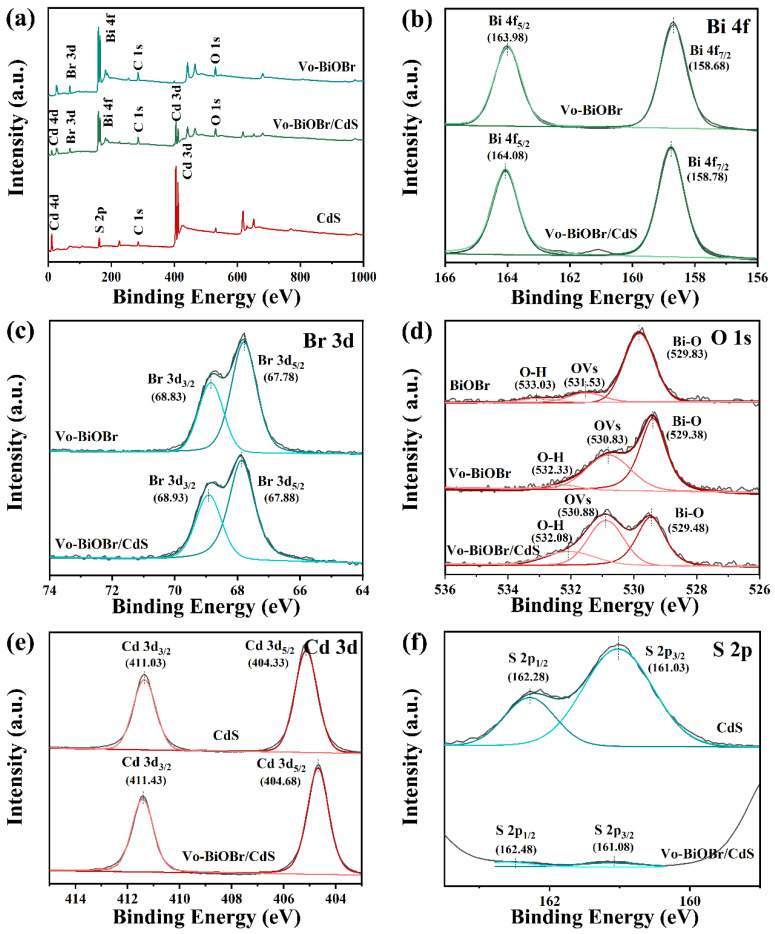
XPS survey spectra of Vo-BiOBr, CdS, and Vo-BiOBr/CdS (**a**); High-resolution XPS spectra of Bi 4f (**b**), Br 3d (**c**), O 1s (**d**), Cd 3d (**e**), and S 2p (**f**) of samples.

**Figure 5 nanomaterials-13-00830-f005:**
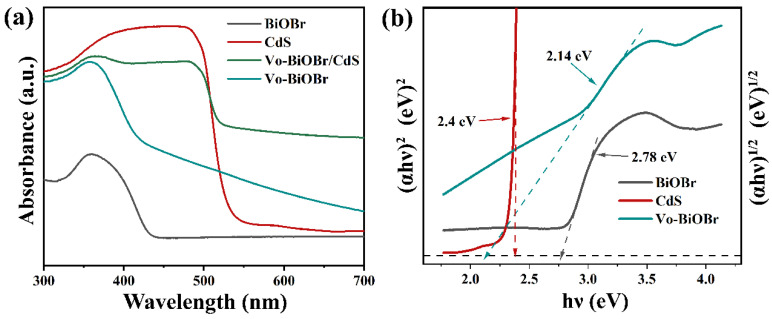
(**a**) UV-vis DRS, and (**b**) Tauc plots of BiOBr, Vo-BiOBr, CdS and Vo-BiOBr/CdS.

**Figure 6 nanomaterials-13-00830-f006:**
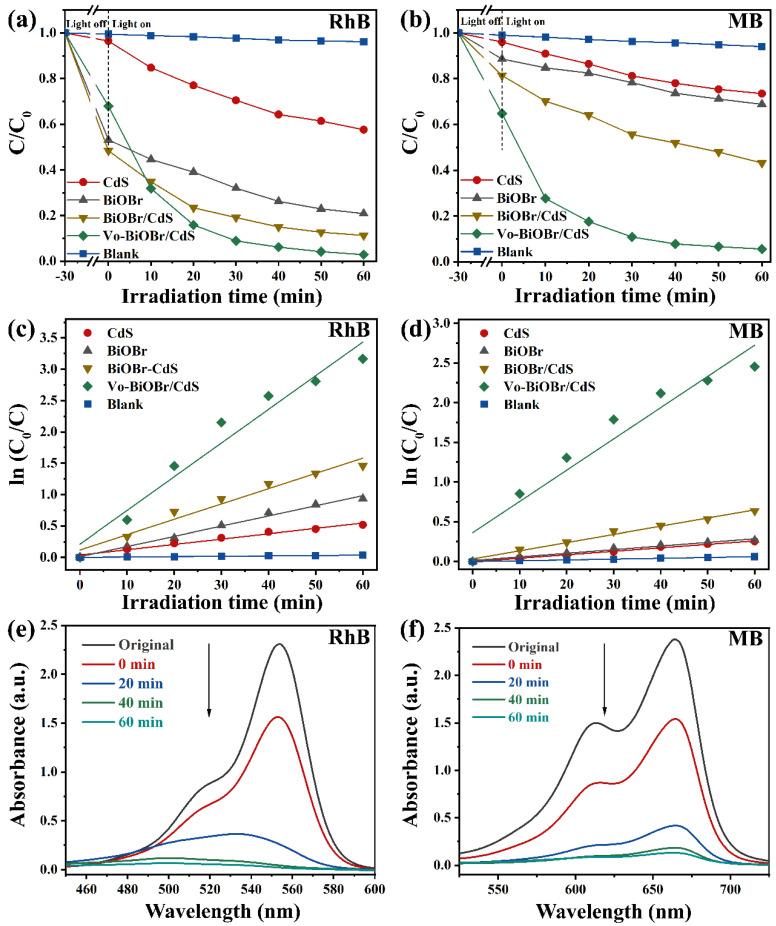
(**a**,**b**) The removal efficiency of RhB and MB over BiOBr, CdS, BiOBr/CdS, and Vo-BiOBr/CdS under the irradiation of LED light; (**c**,**d**) Pseudo-first-order-kinetic fitting curve; (**e**,**f**) Time-dependent absorption spectra of degrading RhB and MB.

**Figure 7 nanomaterials-13-00830-f007:**
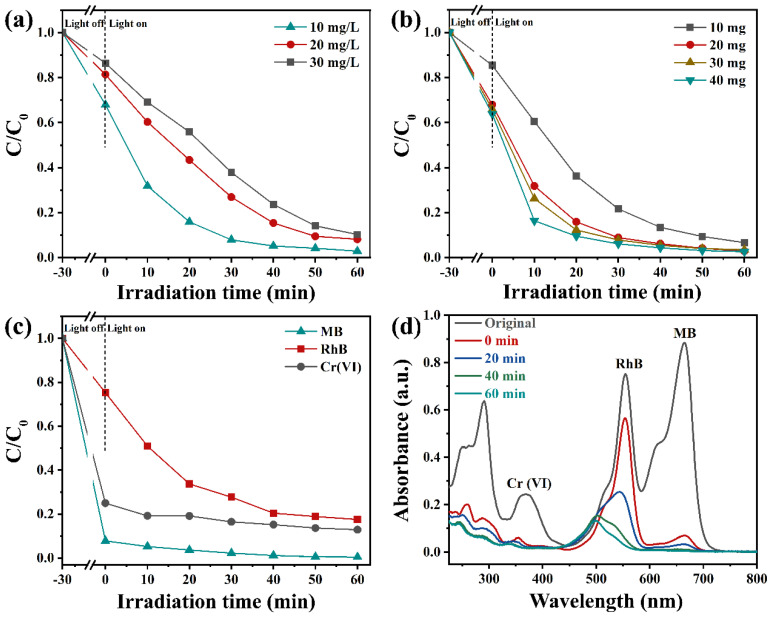
The removal efficiency of RhB solution by various concentrations of RhB (**a**), and various Vo-BiOBr/CdS dosage (**b**). Photodegradation efficiency of RhB-Cr(VI)-MB mixed solution over Vo-BiOBr/CdS (**c**), and time-dependent absorption spectra (**d**).

**Figure 8 nanomaterials-13-00830-f008:**
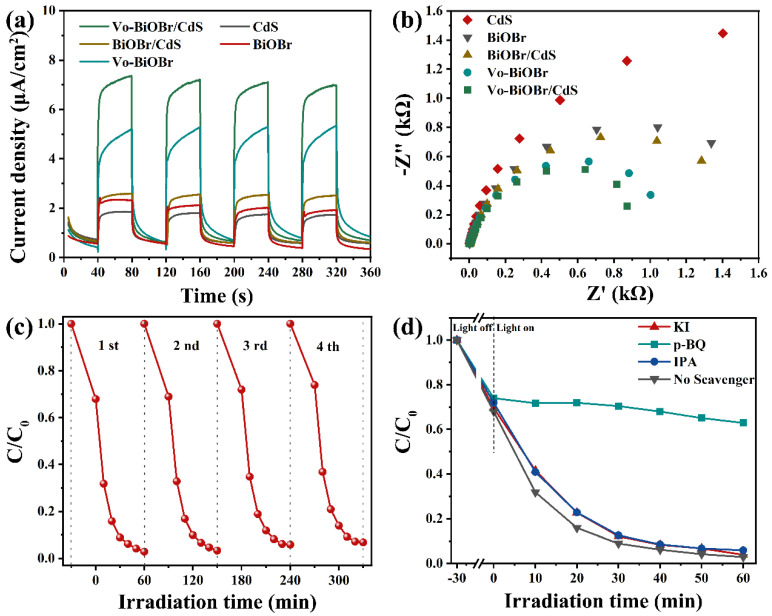
(**a**) Transient photocurrent responses; (**b**) EIS Nyquist plots of photocatalysts; (**c**) cycling test of photocatalytic degrading RhB over Vo-BiOBr/CdS; and (**d**) degradation activity of RhB containing different scavengers.

**Figure 9 nanomaterials-13-00830-f009:**
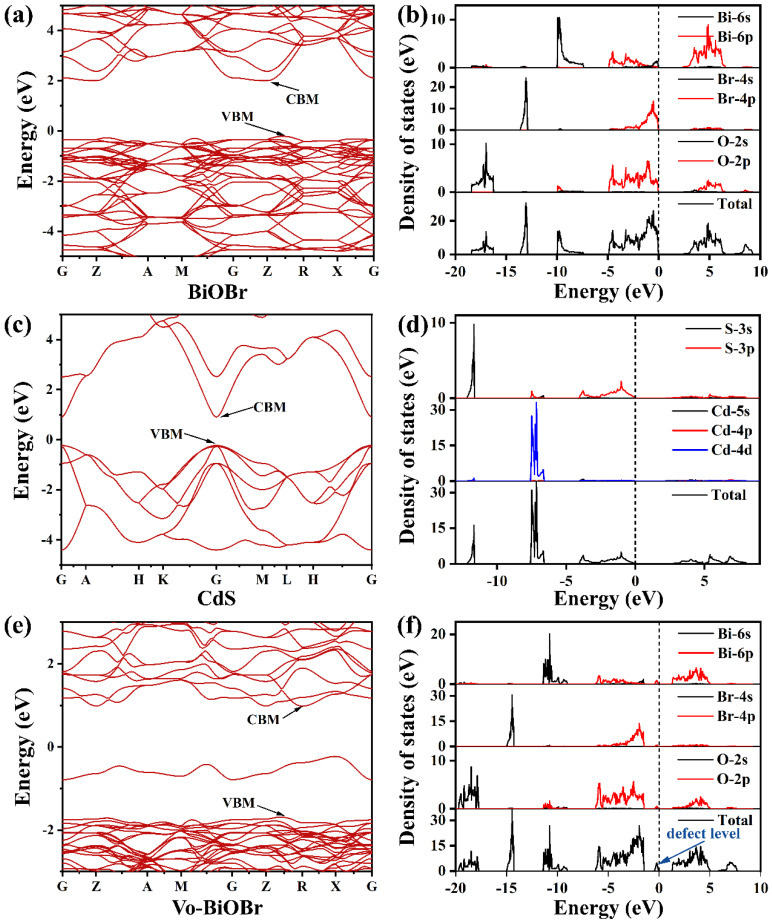
Calculated band structure and density of states of (**a**,**b**) BiOBr, (**c**,**d**) CdS, and (**e**,**f**) Vo-BiOBr.

**Figure 10 nanomaterials-13-00830-f010:**
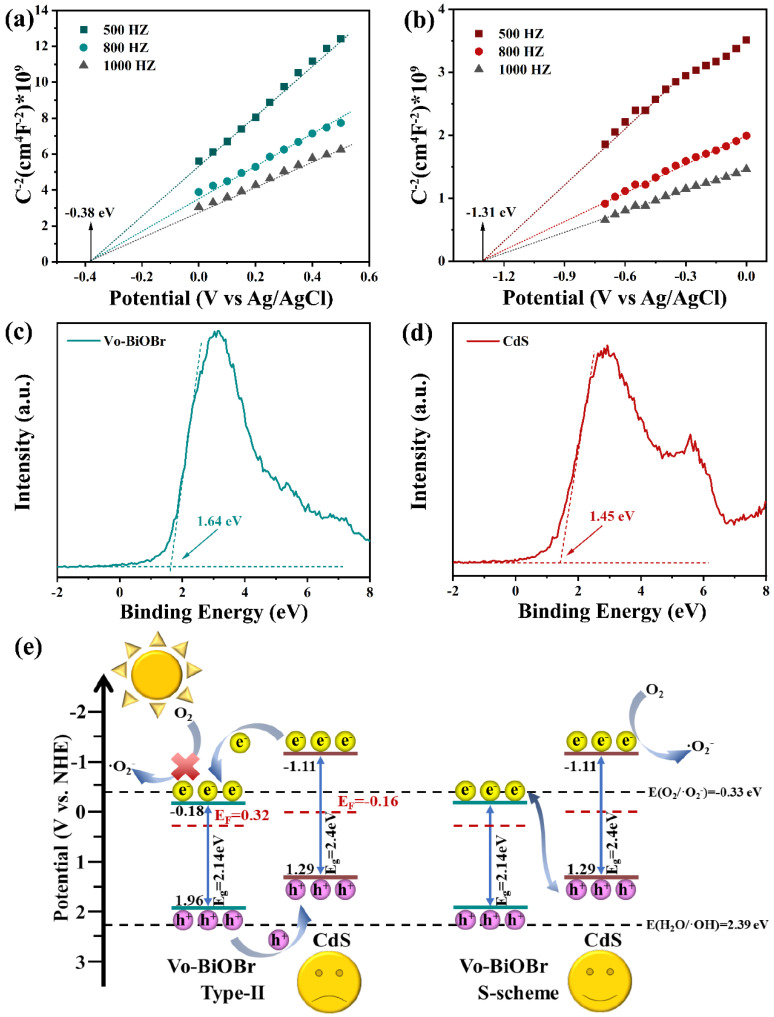
Mott–Schottky spectra of Vo-BiOBr (**a**), and CdS (**b**); XPS valance band spectra of Vo-BiOBr (**c**), and CdS (**d**); band structure of samples and photocatalytic mechanism (**e**).

**Figure 11 nanomaterials-13-00830-f011:**
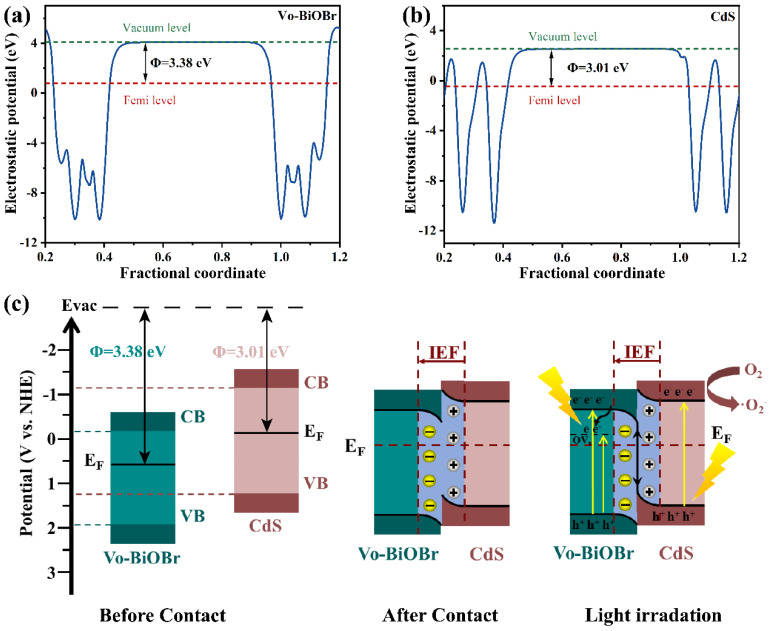
Work functions of (**a**) Vo-BiOBr and (**b**) CdS; Photocatalytic mechanism of Vo-BiOBr/CdS S-scheme heterojunction (**c**).

## Data Availability

All data included in this study are available upon request by contact with the corresponding author.

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
