# Peer review of "Oxygen Vacancies-Rich S-Cheme BiOBr/CdS Heterojunction with Synergetic Effect for Highly Efficient Light Emitting Diode-Driven Pollutants Degradation"

_nanomaterials, 2023, doi:10.3390/nano13050830_

Round 1

Reviewer 1 Report

This paper can be considered for publication  after minor revision:

1. 2.3 Photocatalytic test: line 111, please, define TC

2. Please, check Figure 1( Ethylendia)

3.Please, check line135, 136

4.Line 297  "O2- was decisive active species" for RhB photodegradation. Could you suggest the degradative pathway induced by O2?

Author Response

  1. 2.3 Photocatalytic test: line 111, please, define TC

Response: We appreciate the reviewer’s kind advice. We have defined “TC” as “tetracycline” in the paper. (Line 120, page 3)

  1. Please, check Figure 1(Ethylendia)

Response: Thanks for your valuable comment. The “Ethylendia” in Figure 1 has been modified to “Ethylenediamine”. (Line 141, page 4)

  1. Please, check line135, 136

Response: We are very sorry that we have not expressed it accurately. The sentence was modified as “Because it tended to bind with the unsaturated Bi atom on the BiOBr surface to decrease the surface energy, which leads to the creation of more Vo”. (Line 144, 145, page 4)

  1. Line 297 "·O2-was decisive active species" for RhB photodegradation. Could you suggest the degradative pathway induced by ·O2-?

Response: We sincerely thank the reviewer. Superoxide radicals had strong oxidative activity, and it could oxidize organic matter to carbon dioxide and water. We have suggested the photocatalytic degradative pathway in equation (1-4) according to the comment, as followings: (Line 363, page 14)

Vo-BiOBr/CdS + hv → Vo-BiOBr (e- + h+)/CdS (e- + h+)     (1)

Vo-BiOBr (e-) + CdS (h+) →Vo-BiOBr/CdS (recombination)  (2)

CdS (e-) + O2 →·O2-                                      (3)

  • O2-+ RhB/MB → …→ CO2+ H2O                        (4)

Reviewer 2 Report

The authors have proposed the S-scheme for photocatalysis of RhB and MB dyes using oxygen vacant BiOBr/CdS and shown comparable efficiency. The presentation of data is good. I have only one minor suggestion: Authors must compare their photocatalyst efficiency with the previously reported literature.

Author Response

Authors must compare their photocatalyst efficiency with the previously reported literature.

Response: Thanks for your valuable comment. As suggested by the reviewer, we have compared our photocatalyst efficiency with previously reported literature in the manuscript, such as “Under similar conditions, the Vo-BiOBr/CdS displayed superior photocatalytic performance to that of the reported (Table S2).” (Line 251, page 9)

Table S2 Comparison of the photocatalytic activity of Vo-BiOBr/CdS with other related catalysts reported in the literature.

Catalyst

Experimental conditions Dosage; irradiation time; [pollutant]

Light source

Removal;

Rate (%)

Reference

Vo-BiOBr/CdS

0.5 g/L 60 min

RhB=10 mg/L

MB=20 mg/L

LED lamp

(5 W)

97

94

This work

BiVO4/BiOBr

1 g/L 180 min

RhB=15 mg/L

Xe lamp

(500 W)

90

[1]

BiOBr/ZnO

1 g/L 130 min

RhB=5 mg/L

Xe lamp

(300 W)

95

[2]

BiOCl/BiOBr

1 g/L 360 min

MB=10 mg/L

LED lamp

93

[3]

BiOBr/BiPO4

1 g/L 120 min

RhB=15 mg/L

LED lamp

(12 W)

95

[4]

Bi2MoO6/BiOBr

1 g/L 40 min

MB=20 mg/L

LED lamp

(50 W)

90

[5]

BiOBr-Sn

0.2 g/L 120 min

RhB=10 mg/L

Xe lamp

(500 W)

70

[6]

Reviewer 3 Report

The manuscript deals with synthesis, characterization and photocatalytic properties of composite BiOBr/CdS catalyst using rhodamine B (RhB) and methylene blue (MB) as model pollutants. Topic of manuscript is related to Environmental Chemistry/Water Treatment and fits well to the Journal.

From the technical points of view the manuscript contains practically all standard approaches for characterization of obtained catalysts (X-Ray diffraction, transmission electron microscopy, UV-vis diffuse reflection and photocurrent response). Influence of key factors as catalyst dosage and composition, type and concentration of a pollutant, cycling number were analyzed in details. The synthetized oxygen vacancy-rich BiOBr showed good performance in photooxidation of RhB and MB under visible LED irradiation.

However, I have some technical remarks and questions concerning presentation and discussion of results. Major Revision is recommended according to the remarks below.

Major comments

 1.       In fact, synthesis and mechanism of photocatalysis of BiOBr/CdS catalyst under visible light were repeatedly discussed in the literature (see references below). For, example, according to recent paper of Cui et al. 2017, h+ also can play a role in photocatalytic mechanism. Authors have to discuss recent works in the text and compare their findings with literature ones. Also it will be great to stress advantages of new catalyst with comparison with literature examples.

Yuxi Guo,  Hongwei Huang, Ying He, Na Tian,a   Tierui Zhang, Paul K. Chu,c   Qi An  and  Yihe Zhang, In Situ Crystallization for Fabrication of Core-satellites structured BiOBr-CdS Heterostructure with an Excellent Visible–Light-Responsive Photoreactivity May 2015, Nanoscale 7(27) DOI: 10.1039/C5NR02246K

Haojie Cui, Yawen Zhou, Jinfeng Mei, Zhongyu Li, Song Xu, Chao Yao, Synthesis of CdS/BiOBr nanosheets composites with efficient visible-light photocatalytic activity, 2017, Journal of Physics and Chemistry of Solids 112 DOI: 10.1016/j.jpcs.2017.09.011

Junhua You, Lu Wang, Wanting Bao, Aiguo Yan & Rui Guo  Synthesis and visible-light photocatalytic properties of BiOBr/CdS nanomaterials Journal of Materials Science volume 56, pages6732–6744 (2021)

2.       pH is important parameter which can change the photocatalytic activity. Please, specify pH in photocatalytic experiments and add some discussion about its possible effect in studied system.

3.       O2- radical was proposed as a main intermediate based on experiments with BQ. Is will be great to give additional prove by comparing red-ox potential of O2/O2- pair and RhB/RhB+ cation radical pair and MB/MB+ cation radical pair. Also main reactions demonstrating how O2- can oxidize/reduce studied pollutants should be presented in the text.

 Technical comments:

4.       Lines 46-50 “…showed the highest NH4+ production of 1178 μmol·L-1·g-1·h-1”, “showed the highest CO yield of 88.1 μmol g-1·h-1.” – without any information about studied systems these values have no meaning. Please, omit them, or reformulate these sentences.

5.       Line 52 «However, the improvement of photocatalytic performance by introducing oxygen vacancy is limited.» it is no clear – information about improvement is limited or improvement of photocatalytic performance is not good enough? Please, clarify.

6.       Line 111 and further in the text - “Degradation” is not very good term in relation of Cr6+. It is better to say “reduction”. What is formed upon reduction of Cr6+ ions – is it Cr3+ ions?

7.       TC – what is it? Tetracycline or something else? Please, specify purity and source of all pollutants and radical scavengers used in the study.

8.       Lines 240, 243 Precision of the rate constant (k) is overestimated. Please, round all values to real ones (i.e. 0.05279 to 0.053, 0.04092 to 0.041). The same story with the degradation rate (Line 256) - 95.8% should be 96% and so on.

Author Response

  1. In fact, the synthesis and mechanism of photocatalysis of BiOBr/CdS catalyst under visible light were repeatedly discussed in the literature (see references below). For, example, according to a recent paper of Cui et al. 2017, h+also can play a role in photocatalytic mechanism. Authors have to discuss recent works in the text and compare their findings with literature ones. Also it will be great to stress advantages of new catalyst with comparison with literature examples.

Response: Thanks for your valuable comment. In our work, h+ and ·OH still play a little role in photocatalytic degradation. As suggested by the reviewer, we have read the listed literature. These literature mainly focus on the formation of type II heterojunctions between BiOBr and CdS, while our work mainly focuses on the synergistic effect of oxygen vacancies and S-scheme heterojunctions, and provides a more in-depth discussion of the S-scheme photocatalytic mechanism, which is innovative. Therefore, We have added the following text to illustrate the innovation of this work. For example “In fact, several works on BiOBr/CdS heterojunctions had been reported in recent years. However, little had been reported in the studies on the synergistic effect of Vo and S-scheme heterojunctions to enhance photocatalytic activity.” (Line 62-65, page 2)

In addition, we have compared our photocatalyst efficiency with previously reported literature in the manuscript, such as “Under similar conditions, the Vo-BiOBr/CdS displayed superior photocatalytic performance to that of the reported (Table S2).” (Line 251, page 9)

Table S2 Comparison of the photocatalytic activity of Vo-BiOBr/CdS with other related catalysts reported in the literature.

Catalyst

Experimental conditions Dosage; irradiation time; [pollutant]

Light source

Removal;

Rate (%)

Reference

Vo-BiOBr/CdS

0.5 g/L 60 min

RhB=10 mg/L

MB=20 mg/L

LED lamp

(5 W)

97

94

This work

BiVO4/BiOBr

1 g/L 180 min

RhB=15 mg/L

Xe lamp

(500 W)

90

[1]

BiOBr/ZnO

1 g/L 130 min

RhB=5 mg/L

Xe lamp

(300 W)

95

[2]

BiOCl/BiOBr

1 g/L 360 min

MB=10 mg/L

LED lamp

93

[3]

BiOBr/BiPO4

1 g/L 120 min

RhB=15 mg/L

LED lamp

(12 W)

95

[4]

Bi2MoO6/BiOBr

1 g/L 40 min

MB=20 mg/L

LED lamp

(50 W)

90

[5]

BiOBr-Sn

0.2 g/L 120 min

RhB=10 mg/L

Xe lamp

(500 W)

70

[6]

  1. pH is important parameter which can change the photocatalytic activity. Please, specify pH in photocatalytic experiments and add some discussion about its possible effect in studied system.

Response: We sincerely thank the reviewer’s kind suggestion. We believe that the pH has a significant effect on the photocatalytic degradation rate. However, we have not returned to school so we could not carry out the work. If there is an opportunity in the future, we will study the effect of pH on the photocatalytic degradation rate.

  1. ·O2-radical was proposed as a main intermediate based on experiments with BQ. Is will be great to give additional prove by comparing red-ox potential of O2/O2-pair and RhB/RhB+ cation radical pair and MB/MB+ cation radical pair. Also main reactions demonstrating how ·O2- can oxidize/reduce studied pollutants should be presented in the text.

Response: Thank you for your valuable comments. It is well documented that O2 can be reduced to ·O2-, because the conduction potential of reduction photocatalysts in S-scheme heterojunction is more negative than O2/·O2- according to the reported. For example, Sun et al. determined that superoxide radicals have the greatest effect on photocatalytic processes through trapping agent experiments. In the S-gAP/TAP S-scheme heterojunction system, CB potential of S-gAP was more negative than O2/·O2-, so the ·O2- can be produced (https://doi.org/10.1016/j.jcis.2022.08.089). Moreover, some articles compare the reduction potential of RhB/RhB+ with O2/·O2-.

In addition, superoxide radicals had strong oxidative activity, and it could oxidize organic matter to carbon dioxide and water. We have suggested the photocatalytic degradative pathway in equations (1-4), as followings: (Line 363, page 14).

Vo-BiOBr/CdS + hv → Vo-BiOBr (e- + h+)/CdS (e- + h+)     (1)

Vo-BiOBr (e-) + CdS (h+) →Vo-BiOBr/CdS (recombination)  (2)

CdS (e-) + O2 →·O2-                                      (3)

  • O2-+ RhB/MB → …→ CO2+ H2O                        (4)

  1. Lines 46-50 “…showed the highest NH4+ production of 1178 μmol·L-1·g-1·h-1”, “showed the highest CO yield of 88.1 μmol g-1·h-1.” – without any information about studied systems these values have no meaning. Please, omit them, or reformulate these sentences.

Response: Thank you very much for your kind suggestion. It has been omitted in the manuscript.

  1. Line 52 «However, the improvement of photocatalytic performance by introducing oxygen vacancy is limited.» it is no clear – information about improvement is limited or improvement of photocatalytic performance is not good enough? Please, clarify.

Response: Thank you for your valuable comments. This sentence has been revised as “However, we hope to further improve the photocatalytic performance of the Vo-BiOBr.” according to the comment. (Line 51, page 2)

  1. Line 111 and further in the text - “Degradation” is not very good term in relation of Cr6+. It is better to say “reduction”. What is formed upon reduction of Cr6+ions – is it Cr3+ions?

Response: Thank you for your valuable comments. The sentence included “degradation Cr6+” have been replaced with “Reduction of Cr6+” in the paper, such as “To investigate the degradation and reduction efficiency for the pollutants except for dyes, the photodegradation of TC and photoreduction of Cr (VI) to Cr(III) were executed.”(Line 280, page 10)

According to the previous literatures, Cr6+ will be reduced to Cr3+ through photocatalytic process. For example, “Photocatalytic reduction of aqueous Cr(VI) to harmless trivalent chromium (Cr(III)) has been deemed a feasible and auspicious technique.” (https://doi.org/10.1016/j.seppur.2022.121730)

  1. TC – what is it? Tetracycline or something else? Please, specify purity and source of all pollutants and radical scavengers used in the study.

Response: Thank you very much for your kind suggestion. The TC has been defined as “tetracycline” according to the comment. (Line 119, page 3) Moreover, the purity and source of all pollutants and radical scavengers used in the study have been specified as followings:

“2.1. Materials

Ethylene glycol (EG), cadmium nitrate tetrahydrate (Cd(NO3)2·4H2O), L-Cysteine, ethylenediamine , tetracycline (TC), potassium dichromate(K2Cr2O7) were obtained from Shanghai Aladdin Biochemical Technology Co., Ltd (analytically pure, 99%, Shanghai, China). polyvinyl pyrrolidone (PVP), isopropanol, potassium iodide and p-benzoquinone were obtained from the Macklin company (analytically pure, 99%, Shanghai, China). bismuth nitrate pentahydrate (Bi(NO3)3·5H2O), potassium bromide (KBr), MB and RhB were purchased from Tianjin Kwangfu Fine Chemical Research Institute (analytically pure, 99%, Tianjin, China).”. (Line 73-81, page 2)

  1. Lines 240, 243 Precision of the rate constant (k) is overestimated. Please, round all values to real ones (i.e. 0.05279 to 0.053, 0.04092 to 0.041). The same story with the degradation rate (Line 256) - 95.8% should be 96% and so on.

Response: Thank you for your valuable comments. The rate constant have been revised as “0.053 min-1, 0.041 min-1” and “96%, 90%, and 86%” in the paper. (Line 247, 253, 258, page 9)

Round 2

Reviewer 3 Report

I am generally satisfied by revised manuscript and now it worth to be published in the Journal